# Frequency Division Multiplexing of Terahertz Waves Realized by Diffractive Optical Elements

Paweł Komorowski [1,*] , Patrycja Czerwińska [2] , Mateusz Kaluza [2] , Mateusz Surma [2] ,
Przemysław Zagrajek [3] , Artur Sobczyk [2] , Wiesław Ciurapiński [3] , Ryszard Piramidowicz [1]
and Agnieszka Siemion [2]

1 Institute of Microelectronics and Optoelectronics, Warsaw University of Technology, Koszykowa 75, 00662 Warsaw, Poland; ryszard.piramidowicz@pw.edu.pl
2 Faculty of Physics, Warsaw University of Technology, Koszykowa 75, 00662 Warsaw, Poland; 01121203@pw.edu.pl (P.C.); mateusz.kaluza.dokt@pw.edu.pl (M.K.); mateusz.surma.dokt@pw.edu.pl (M.S.); artur.sobczyk@pw.edu.pl (A.S.); agnieszka.siemion@pw.edu.pl (A.S.)
3 Institute of Optoelectronics, Military University of Technology, Kaliskiego 2, 00908 Warsaw, Poland; przemyslaw.zagrajek@wat.edu.pl (P.Z.); wieslaw.ciurapinski@wat.edu.pl (W.C.)
* Correspondence: p.komorowski@imio.pw.edu.pl

**Abstract:** Recently, one of the most commonly discussed applications of terahertz radiation is wireless telecommunication. It is believed that the future 6G systems will utilize this frequency range. Although the exact technology of future telecommunication systems is not yet known, it is certain that methods for increasing their bandwidth should be investigated in advance. In this paper, we present the diffractive optical elements for the frequency division multiplexing of terahertz waves. The structures have been designed as a combination of a binary phase grating and a converging diffractive lens. The grating allows for differentiating the frequencies, while the lens assures separation and focusing at the finite distance. Designed structures have been manufactured from polyamide PA12 using the SLS 3D printer and verified experimentally. Simulations and experimental results are shown for different focal lengths. Moreover, parallel data transmission is shown for two channels of different carrier frequencies propagating in the same optical path. The designed structure allowed for detecting both signals independently without observable crosstalk. The proposed diffractive elements can work in a wide range of terahertz and sub-terahertz frequencies, depending on the design assumptions. Therefore, they can be considered as an appealing solution, regardless of the band finally used by the future telecommunication systems.

**Keywords:** diffractive optical elements; THz; 3D printing; multiplexing; frequency division multiplexing; telecommunication

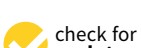

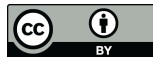

## 1. Introduction

Recently, the rapid development of terahertz (THz) technologies is observed [1]. Unique features of the THz waves allowed them to find applications in many fields of science [2,3], nondestructive testing [4–7], medical diagnostics [8–10] as well as telecommunication [11–14]. Although the precise technologies and bands reserved for future wireless communication systems are not yet known, it is commonly believed that they will utilize terahertz frequencies [15]. One of the crucial aspects of all modern telecommunication systems is the constant demand for increasing bandwidth. It can be met by systems, in which more than one communication link can be placed in a single communication medium (which is known as multiplexing). The multiplexing methods are generally based on the launching of many signals into a single channel and their subsequent separation at the detector's side. To successfully demultiplex independently propagating signals, one must separate them in some dimension or feature space. The simplest method is to distribute signals in the space, which is known as Space Division Multiplexing (SDM) [16]. In the case of

wireless data transmission, it is often implemented in the form of so-called MIMO systems (Multiple Input Multiple Output) [17]. Multiplexing techniques based on signal division in time, polarization, modal distribution, or orbital angular momentum have also been shown and implemented in many telecommunication systems [18–20]. However, probably, the most widespread and impactful multiplexing method is based on the separation of the wavelength of electromagnetic waves. In optical fiber communication systems, it is known as Wavelength Division Multiplexing (WDM), while, in wireless transmission, the more common name is Frequency Division Multiplexing (FDM) [21]. Recently, more and more multiplexing and demultiplexing solutions are also proposed for the THz band. The natural approach is to adapt solutions known from other telecommunication systems. Therefore, the most commonly considered multiplexing methods in the THz spectral range are based on the MIMO systems [22–25]. However, the first THz multiplexing solutions based on frequency division [26,27], polarization division [26], space division [28], or orbit angular momentum division [29] are also reported. In this paper, we present a novel method of frequency division multiplexing of the THz radiation, based on the combination of two different diffractive optical elements (DOEs). Designed structures separate spatially signals coming from a single data transmission channel with respect to the carrier frequency.

DOEs are optical elements that introduce a defined change in the complex amplitude of the incident radiation to obtain the intended intensity distribution in the output plane [30]. A particular case is a phase kinoform that progressively changes the phase from 0 to $2\pi$ and can redirect the whole radiation into a single order of diffraction [31]. According to the theory, such phase coding allows for obtaining 100% diffraction efficiency of the structure.

The idea presented in this article is to combine a diffraction grating with the diffractive converging lens into a single phase DOE coded in the form of kinoform. A diffraction grating illuminated by the monochromatic radiation redirects it into $m$ diffraction orders, according to the well-known equation:

$$m\lambda = d \sin \alpha_m, \tag{1}$$

where $\lambda$ is the design wavelength (DWL), $m$ is considered diffraction order, $d$ is grating constant, and $\alpha_m$ is an angle between the 0th (main optical axis) and $m$th order of diffraction. One can easily notice that the change of the wavelength $\lambda$ (that is inversely proportional to frequency) introduces a change in the diffraction angle $\alpha_m$ of all orders of diffraction except 0th. It means that the radiation of longer wavelength (smaller frequency) will be redirected farther from the optical axis within the single diffraction order. The shifts will be more significant for higher orders of diffraction, however at the cost of the strongly diminished diffraction efficiency. Therefore, we decided to focus on the 1st order of diffraction. In such an approach, all other diffraction lines should be considered as a source of power losses, which can be reduced with the proper design of diffraction grating. Binary phase grating with 0-$\pi$ phase jump forms only odd orders of diffraction and redirects 40.5%, 4.5%, and 1.6% of the incoming power to the 1st, 3rd, and 5th orders of diffraction, respectively. Separation of the diffraction orders of the diffraction grating is observed in the far field (Fraunhofer region), which in the case of the THz wavelengths and aperture close to 100 mm lies several meters behind the structure. To overcome this issue and minimize the setup, an additional converging lens has been combined with the grating. In such a case, the diffraction orders are separated in the focal plane of the lens. Application of the DOEs is also advantageous, due to simple and cost-efficient manufacturing methods. It turns out that some polymers, e.g., Nylon 12 (PA12), Polypropylene (PP), or High Impact Polystyrene (HIPS), have a low absorption coefficient and refractive index in the range of 1.5 to 1.6 in the THz band [30,32]. Thus, those materials can be used in the manufacturing process of optical structures for the THz spectral range. The lenses can be fabricated from filament, powder, or resin using the respective 3D printing methods (additive manufacturing methods): Fused Deposition Modeling (FDM*) [33–35], Selective Laser Sintering (SLS) [35,36], and Digital Light Processing (DLP) [35,37]. Additive manufacturing in many cases can guarantee high

resolution and good quality of diffractive optical elements (DOEs) as well as a low cost of the production process [35].

In this paper, we present four DOEs, designed for four different focal lengths of 100 mm, 200 mm, 300 mm, and 400 mm. All structures have been designed for the DWL of 1 mm (which corresponds to the frequency of 300 GHz). All structures were simulated for the DWL as well as for other wavelengths. The structures with focal lengths of 200 mm and 400 mm have been chosen for experimental evaluation. All structures have been manufactured by means of SLS additive manufacturing from Nylon 12 (PA12). Design, modeling, and manufacturing of the DOEs are more precisely described in the next section.

## 2. Materials and Methods

### 2.1. Design

All structures have been designed as a combination of a convergent lens and a diffraction grating. Phase distribution ($\theta$) of the lens has been defined according to the thin off-axis lens equation:

$$\theta = -\frac{2\pi}{\lambda}\sqrt{r^2 + f^2}, \tag{2}$$

where $\lambda$ is the design wavelength (DWL), $r$ is the radial coordinate in the lens plane (distance from the center of the structure), and $f$ denotes the focal length. Then, the according transmittance (T = exp($i\theta$)) has been multiplied by the transmittance of the binary phase grating with a constant $d = 1.872$ mm (introducing subsequent phase shifts of 0 and $\pi$). The transmittance function of such a grating consists of vertical stripes with the width of 0.936 mm and values alternating between 1 and $-1$. For the convenience of numerical simulations and manufacturing, the analytical equation has been sampled with the 0.117 µm pitch. Four structures with focal lengths varying from $f = 100$ mm through $f = 200$ mm and $f = 300$ mm to $f = 400$ mm have been designed for the DWL approximately equal to 1 mm (which corresponds to the frequency of 300 GHz). Phase distributions of all designed lenses are illustrated in Figure 1.

### 2.2. Numerical Simulations

The performance of the structures has been simulated using a convolution based light propagation algorithm [38] implemented in Wolfram Mathematica. It emulates the propagation of radiation by convolving the complex amplitude of the optical field with the impulse response function of the free space. For the two structures with shorter focal lengths ($f = 100$ mm and $f = 200$ mm), calculations have been performed on $4096 \times 4096$ matrices of 0.117 µm square pixels. For the other two structures, the matrix size has been increased to $8192 \times 8192$ due to the higher perpendicular separation of the focal spots for longer propagation distances. Firstly, all structures have been verified for the design frequency of 300 GHz. Next, the frequency of the incoming radiation has been blue- and red-shifted with the step equal to 30 GHz. Simulated positions and dimensions of the focal spots for the two structures with shorter focal lengths are illustrated in the form of intensity distributions in Figure 2.

These distributions present fragments of the focal planes cut in such a way that the optical axis is placed in the middle of the left edge. The 0th order of diffraction can be seen in the form of dots and rings around the optical axis. For the DWL, it is nonexistent. However, the further the frequency of illuminating radiation is shifted, the stronger non-diffracted signal is observed. As expected, the longer wavelengths are focused farther from the optical axis into more distorted spots. Two effects contribute to this distortion. Firstly, the diffraction limit of light (defined as the diameter of Airy spot) depends linearly on the wavelength, hence the widening of the spots for longer wavelengths. Secondly, for the off-axis focusing, the circularity of the focal spot decreases approximately linearly with the tangent of the beam deflection angle [39], hence the horizontal elongation of the focal spots. It can be noticed that, in the case of the $f = 100$ mm structure, the spots obtained for the frequencies differing by 30 GHz are hardly separable. For the $f = 200$ mm structure,

the separation is better, especially in the shorter wavelength division. Focal spots for longer wavelengths are still overlapping but can be easily distinguished. It shows that longer focal lengths should provide better separation of the signals of different frequencies, which has been verified for the $f = 300$ mm and $f = 400$ mm structures in Figure 3.

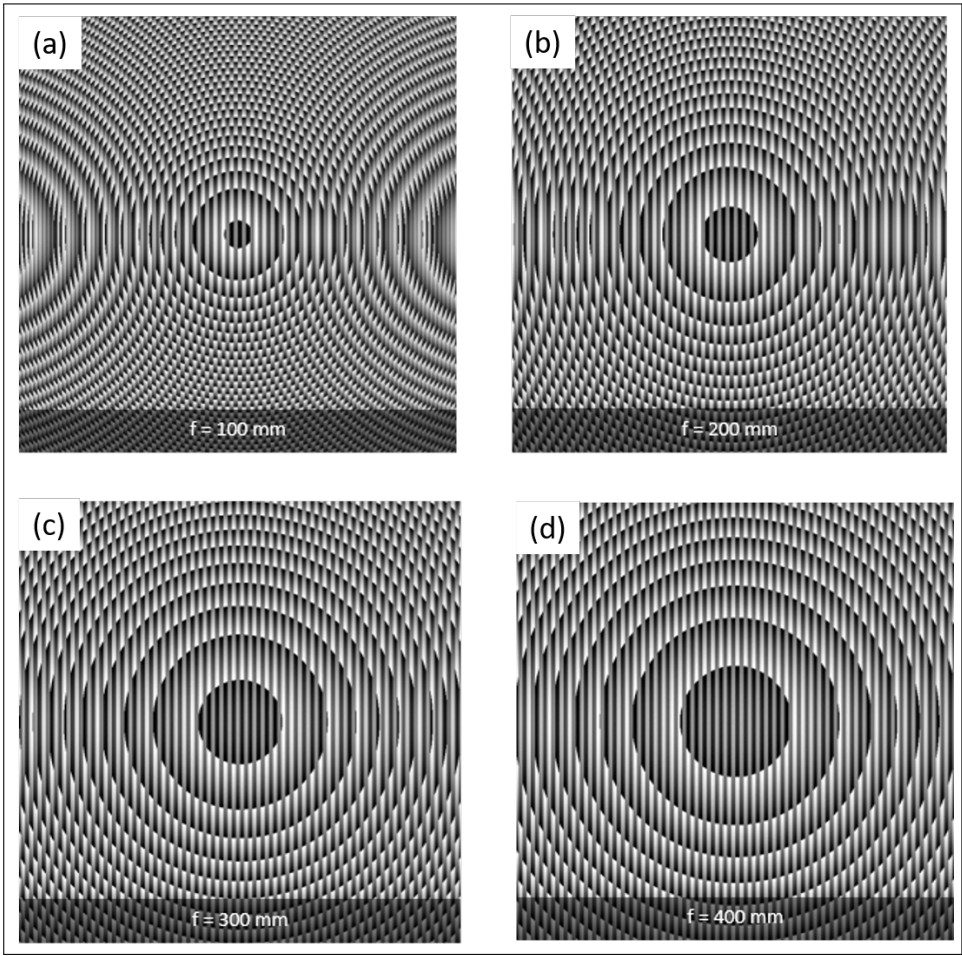

**Figure 1.** Phase distributions of four diffractive structures with different focal lengths: (**a**) $f = 100$ mm; (**b**) $f = 200$ mm; (**c**) $f = 300$ mm; and (**d**) $f = 400$ mm. White color corresponds to $2\pi$ phase shift, while a black to no phase shift is introduced by the structure.

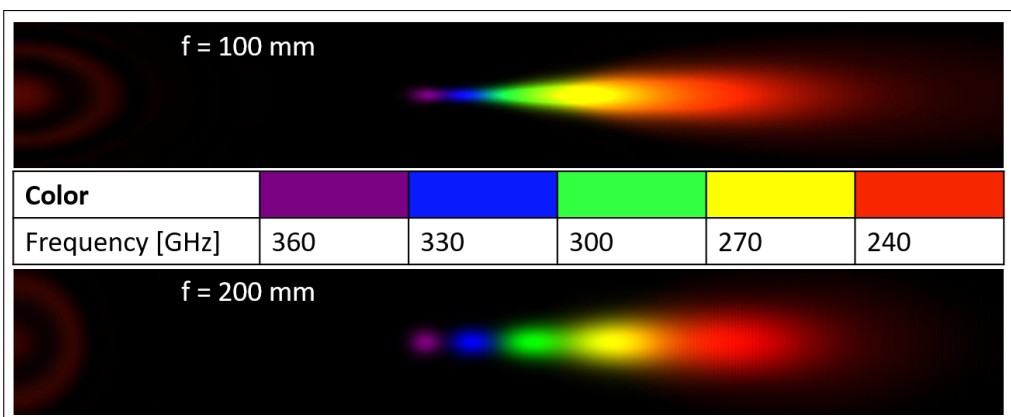

**Figure 2.** Calculated positions of the focal spots for structures with the focal lengths $f = 100$ mm (**top**) and $f = 200$ mm (**bottom**). Intensity distributions are shown with a main optical axis in the middle of left edge of the illustrated area. Colors denote the frequency of the illuminating radiation as in the legend.

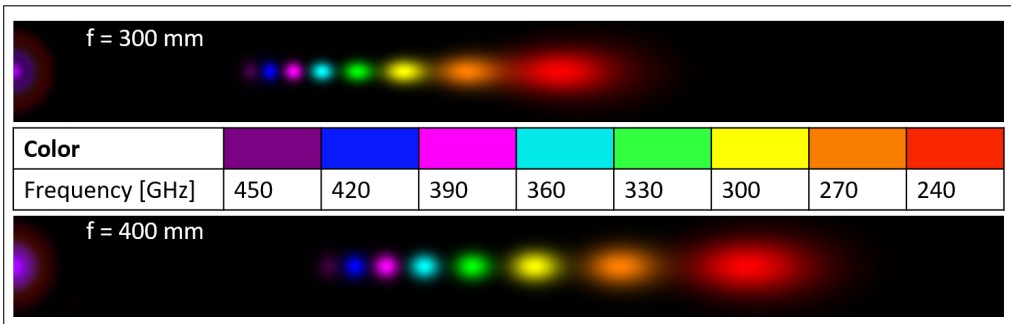

**Figure 3.** Calculated positions of the focal spots for structures with the focal lengths $f = 300$ mm (**top**) and $f = 400$ mm (**bottom**). Intensity distributions are shown with the main optical axis in the middle of left edge of the illustrated area. Colors denote the frequency of the illuminating radiation as in the legend.

It can be noticed that, at these focal distances, the signals spaced by 30 GHz are clearly spatially separated. Moreover, radiation of higher frequency can be demultiplexed by the designed DOEs (up to 450 GHz, which corresponds to a 150 GHz shift from the DWL).

### 2.3. Manufacturing

Phase distributions of all structures have been designed accordingly to the previously described matrices in the form of gray-scale images presented in Figure 1. The size of the matrices as well as the pixel dimensions allowed for creating 3D models corresponding to the 2D bitmaps from simulations. The value of each pixel of the matrix is an integer in the range from 0 to 255, representing different tones of gray. The bitmaps have been extruded with respect to the pixel values. The real thickness $h$ (dependent on the refractive index $n$ of the material used for manufacturing) of each structure is proportional to the introduced phase delay $\phi$ and can be calculated from the following formula:

$$h(x,y) = \frac{\lambda}{n-1} \frac{\phi(x,y)}{2\pi}. \tag{3}$$

Hereby, the created models had the dimensions of 74.06 mm × 74.06 mm × 1.67 mm. An additional thin layer of a substrate has been added to the model to provide stiffness of the structure. Manufactured DOEs have also been surrounded with the 10 mm thick frame for easier handling and damage protection. The substrate is flat, therefore it introduces a constant phase delay to the whole structure which does not influence its behavior (apart from slightly increased absorption loss). The total structure size is 94.06 mm × 94.06 mm × 8.00 mm (this thickness results from the frame size, not the structure itself).

Two structures have been manufactured using Selective Laser Sintering. This method is based on sintering of a polymer in the form of powder by laser light [35,36] with a high vertical resolution of 0.1 mm. The structures have been manufactured from PA2200 (powder form of PA12). This material is commonly used in 3D printing and is characterized with above average optical properties in the THz spectral range [30,33,40]. The measurements performed using the THz Time Domain Spectroscopy (THz-TDS) method have shown that PA12 is characterized by a refractive index of 1.56 (at 300 GHz) accompanied by a low absorption coefficient of 0.9 cm$^{-1}$ that corresponds to high transparency [41]. DOEs with focal lengths $f = 200$ mm and $f = 400$ mm have been selected for the experimental evaluation. The manufactured structures can be seen in Figure 4.

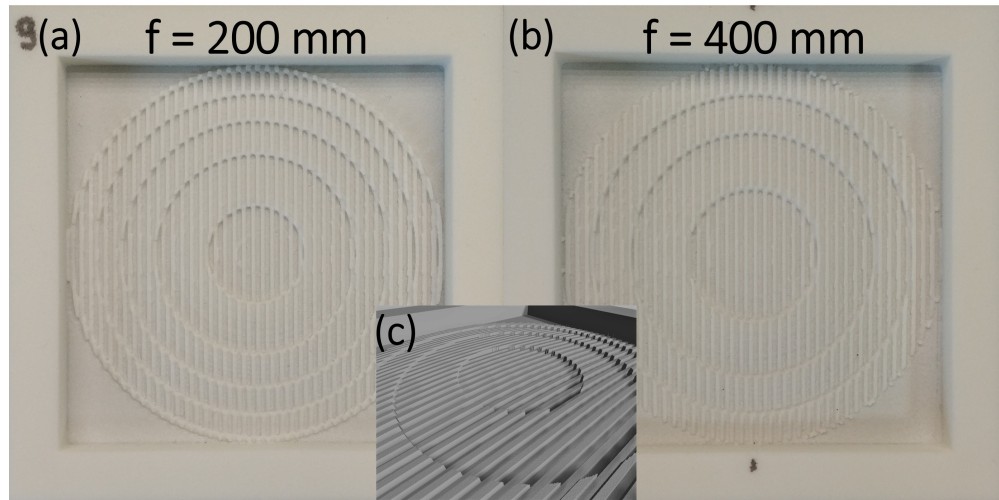

**Figure 4.** The photographs of DOEs with (**a**) $f = 200$ mm and (**b**) $f = 400$ mm manufactured by the SLS method from PA2200 (polyamide 12). The inset (**c**) shows the fragment of the 3D visualization of the $f = 400$ mm structure.

## 3. Results

### 3.1. Experimental Setup

The scheme of the setup used for the experimental verification of the DOEs is shown in Figure 5.

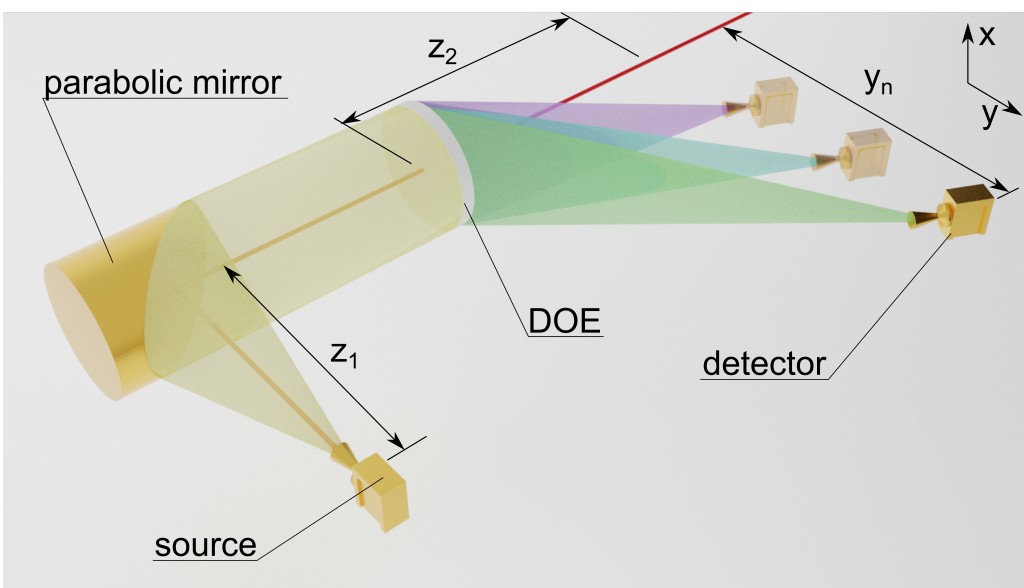

**Figure 5.** Image depicting experimental setup. $z_1$ was equal to 600 mm, $z_2$ was established as optimal distance for 300 GHz for $f = 200$ mm and $f = 400$ mm separately. Distance of the scan in the measurement plane from optical axis $y_n$ depended on frequency of reconstruction (details in Table 1).

Schottky diode based frequency multiplier with antenna (produced by Virginia Diodes Inc., Charlottesville, VA, USA—VDI) has been used as a source of terahertz radiation. The DOEs under measurement have been illuminated by a quasi-plane wave, collimated with a parabolic mirror. Intensities of the focal spots have been measured with a VDI Schottky diode with a hemispherical lens attached to it, placed on the motorized stage. All scans have been performed in the plane perpendicular to the optical axis in the 10 mm × 10 mm matrices with a 1 mm step. The detector used has an angle of acceptance limited to about 15–20°. The radiation with the frequency of 300 GHz is redirected at an angle of 32 ° (according to Equation (1)). Therefore, to obtain the best possible information about the

intensity and shape of the focal spots, it is advisable to rotate the detector in a way that it faces the middle of the DOE. All measurements have been performed with the detector aligned to the direction of propagation of the 300 GHz beam. The influence of the directivity and rotation angle of the detector has been discussed more specifically in our previous publications [41]. Each DOE has been tested with multiple frequencies according to their design: 240 GHz, 270 GHz, 300 GHz, 330 GHz, 360 GHz and additionally 390 GHz for DOE with $f = 400$ mm. The distance between DOE and the measurement plane ($z_2$) has been adjusted for the 300 GHz beam by maximizing the measured intensity in the scan performed along the optical axis and kept constant for a given DOE. The angle of the detector has been optimized for the 300 GHz beam and kept unchanged for a given focal length for the sake of comparison clarity. The distance $z_2$ for DOEs has been equal to 179 mm for $f = 200$ mm and 235 mm for $f = 400$ mm. The difference between the designed distances and the registered ones was caused by illuminating the structures by quasi-plane waves, which turned out to be slowly convergent.

### 3.2. Focal Spots

All focal spots have been scanned separately and then inscribed into the common $xy$ plane. Empty points have been filled with zeros (black color). The registered intensity distributions for the DOEs with focal lengths $f = 200$ mm and $f = 400$ mm can be seen in Figure 6.

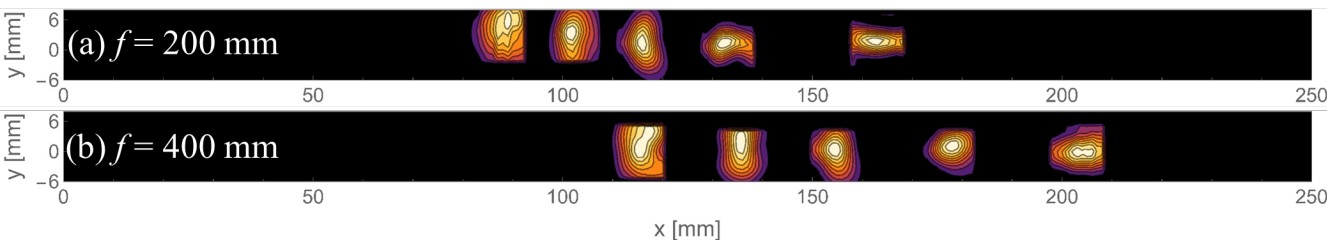

**Figure 6.** Intensity distributions of the focal spots measured for different radiation frequencies inscribed into the common $xy$ diagrams illustrating focal plane for the focal length (**a**) $f$ = 200 mm and (**b**) $f$ = 400 mm.

First of all, it should be noticed that the focal spots for different frequencies are shifted, as predicted in the simulations. Longer focusing distance also translates into better separation of the focal spots. Although the precise positions of the particular spots differ slightly from the simulated ones, they still remain proportional to them. These positions depend on the focusing distance, which on the other hand comes from the convergence of the illuminating beam. This dependence has been analyzed in Table 1.

**Table 1.** Positions of shifted focal spots in the simulation and experiment for each DOE, corresponding to particular source frequencies. For each focal spot, the ratio between the calculated and measured shift value is given.

| Frequency [GHz] | 240 | 270 | 300 | 330 | 360 | 390 |
|---|---|---|---|---|---|---|
| Focal spot position [mm] ($f$ = 200 mm) | | | | | | |
| Simulation | 177 | 145 | 126 | 109 | 95 | - |
| Experiment | 162 | 132 | 115 | 102 | 89 | - |
| Ratio | 0.92 | 0.91 | 0.91 | 0.94 | 0.94 | - |
| Focal spot position [mm] ($f$ = 400 mm) | | | | | | |
| Simulation | 356 | 304 | 260 | 220 | 204 | 176 |
| Experiment | 204 | 177 | 154 | 132 | 116 | 101 |
| Ratio | 0.57 | 0.58 | 0.59 | 0.60 | 0.57 | 0.57 |

The table presents the comparison of the measured and simulated positions of each focal spot for both DOEs. It is clearly seen that the experimental values differ from those obtained in the simulations. It should be noted, however, that the ratios of the shifts from the optical axis obtained in the experiment and in the simulation are very consistent across the focal spots of the given structure. For $f = 200$ mm, the shift ratios vary from 0.91 to 0.94, while the ratio of the experimental focusing distance to the focal length was equal to 0.90. Thereby, the shift ratios are close to each other and to the proportion of the experimental focusing distance and theoretical focal length. It proves that the redirection angles are constant for given frequencies even for the illumination with the radiation deviating from the plane wave. This observation is very important from the perspective of the potential application, as the structures are not limited by the convergence of the illuminating beam. The same behavior can be observed for $f = 400$ mm. The ratio between the experimental focusing distance and the focal length has been equal to 0.59. In this case, the shift ratios of focal spots have been similar to this value, varying from 0.57 to 0.60.

Precise scans of intensity distributions in the focal spots are presented in Figure 7.

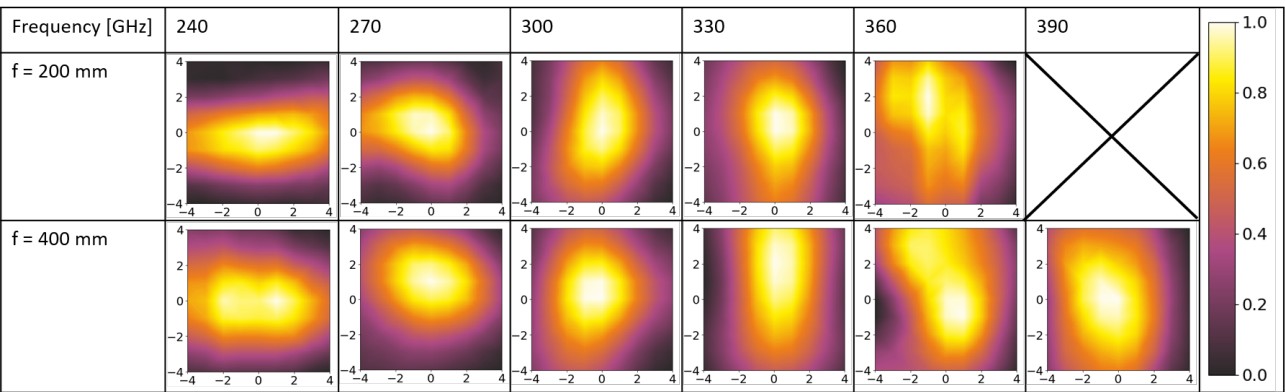

**Figure 7.** Intensity distributions of focal spots obtained for range of frequencies for DOEs with $f = 200$ mm and $f = 400$ mm. Presented intensities were cropped to size of 4 mm × 4 mm to show the shape of focal spots. Each intensity distribution was normalized to the maximum value registered while scanning a particular focal spot. Power emitted from the source depends on the frequency; therefore, intensities of the focal spots cannot be compared directly.

The most circular and the smallest focal spots have been registered for the 300 GHz frequency. This result is expected, as both the structure and experimental procedure have been optimized for this frequency. The geometrical parameters of the focal spots become, however, slightly worse for increasing values of the frequency shift from the DWL. It is connected both with the design of the DOE and the experimental procedure. Firstly, radiation with the frequency differing from the DWL is focused at different distances. As stated before, all measurements have been performed in the single $xy$ plane; therefore, the scans of all frequencies except 300 GHz have been gathered not exactly in their focal planes. Nevertheless, it must be noted that, for these geometrical parameters and wavelengths, the focal spots are relatively long (high Rayleigh length) in comparison with the focal length shifts. The second thing is the rotation angle of the detector (optimized for 300 GHz). The farther the detector is moved from the original position, the higher is the angle between the normal of the scanning plane and the propagation direction of the THz beam. In such a case, the cross-section of the circular beam becomes elliptical, which can be observed in Figure 7. This effect is also stronger for smaller frequencies (higher shifts from the optical axis), as described in the simulations section.

### 3.3. Multiplexing with Frequency Division

Measurements presented in the previous subsection have been supplemented with a simple parallel data transmission experiment. Results shown above prove that the THz radiation with different frequencies is focused and redirected at different angles by the designed structures. Therefore, they can be used to multiplex THz channels in the frequency division. To visualize such functionality, THz radiation with two different carrier frequencies has been coupled into a common optical path by a semitransparent mirror (beam splitter). Combined signals illuminated the designed multiplexing DOE, where the signals have been spatially separated. The photograph of the optical setup with the beam paths marked with red and blue lines is presented in Figure 8.

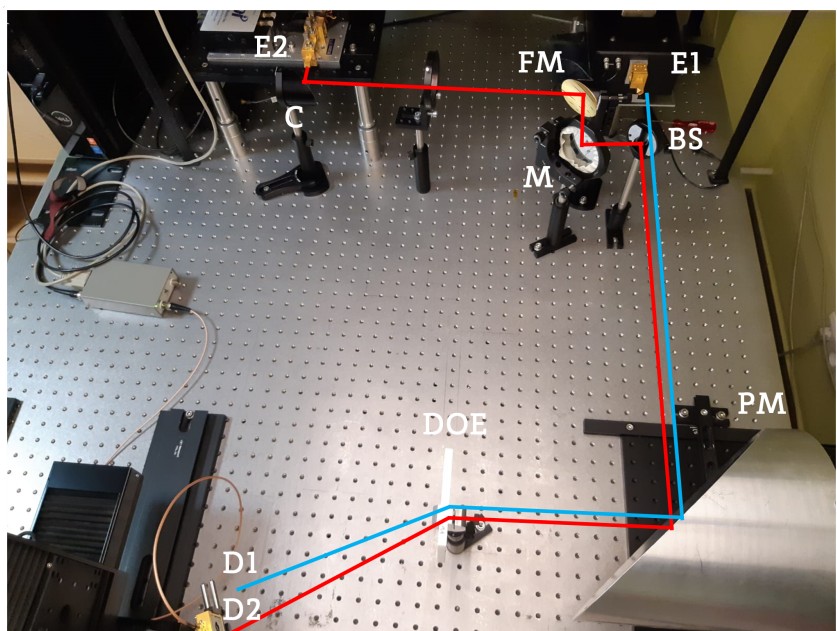

**Figure 8.** The optical setup used for verification of the spatial separation of signals with frequencies of 300 GHz and 330 GHz. E1 and E2—emitters, C—collimator, FM—focusing mirror, M—flat mirror, BS—beam splitter, PM—parabolic mirror, DOE—designed structure, D1 and D2—detector positions.

Two independent signals with the frequencies of $\nu_1 = 300$ GHz and $\nu_2 = 330$ GHz have been modulated with the frequencies of $\omega_1 = 6$ Hz and $\omega_2 = 24$ Hz, respectively. In the common path (before DOE), two separate signals and their combination have been measured. Results are presented in Figure 9 in the form of time domain waveforms as well as the corresponding Fourier transforms.

Blue and orange waveforms correspond to the $\nu_1$ and $\nu_2$ signals, respectively, and have been measured with only a single emitter turned on. The green waveform, on the other hand, has been observed with both transmitters operating. It can be seen that the broadband detector integrates both signals and the obtained waveform is a sum of both modulation frequencies. It can be observed even more clearly in the Fourier spectra of the recorded waveforms, where both modulation frequencies appear in the joined power spectral density (PSD) dependence. Higher modulation frequencies visible for the 300 GHz beam (with $\omega_1 = 6$ Hz) are a numerical artifact connected with the finite length of the transformed time domain waveform resulting in the presence of higher harmonics of the original modulation frequency $\omega_1$.

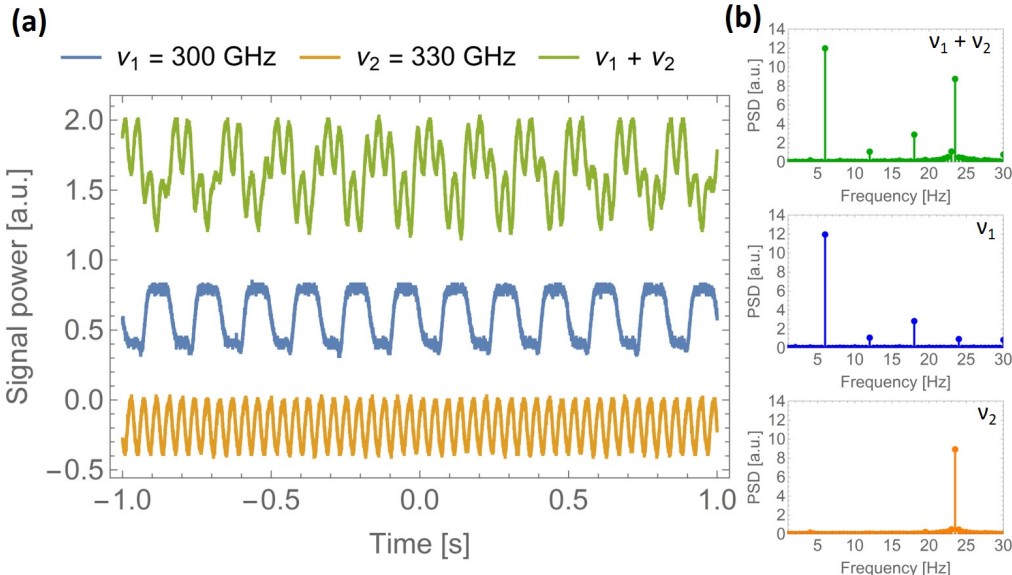

**Figure 9.** (**a**) Reference results of signals registered in the common path (before DOE). The intensity registered in time domain for frequencies of $v_1 = 300$ GHz and $v_2 = 330$ GHz with rectangular modulation of $\omega_1 = 6$ Hz and $\omega_2 = 24$ Hz measured in the single optical path before signals separation—both sources are on (green), only a 300 GHz source is on (blue) and only a 330 GHz source is on (orange); (**b**) the power spectral densities for modulated signal frequencies $v_1$ and $v_2$ obtained by Fourier Transform.

Analogous measurements have also been performed in the positions D1 and D2 marked in Figure 8. It is assumed that $v_1$ frequency should be focused in the D1 position, while $v_2$ in D2. Results are presented in Figure 10.

In all charts, the blue, orange, and green lines denote, respectively, signals with the $v_1$, $v_2$, and both frequencies combined. It can be noticed that, in the D1 position, only the $\omega_1$ modulation frequency is detected, while, in the D2 position, only the $\omega_2$ modulation frequency is registered. In both cases, when only the opposing transmitter is turned on, no modulation can be observed in the time domain waveform. Corresponding Fourier transforms also prove it, as the corresponding power spectral distributions are flat. This experiment, although simple, shows that the designed structures can be successfully used for simultaneous data transmission of more than one signal within a common optical path. Multiplexing and demultiplexing of signals differing in the carrier frequency can be performed with the proposed DOEs.

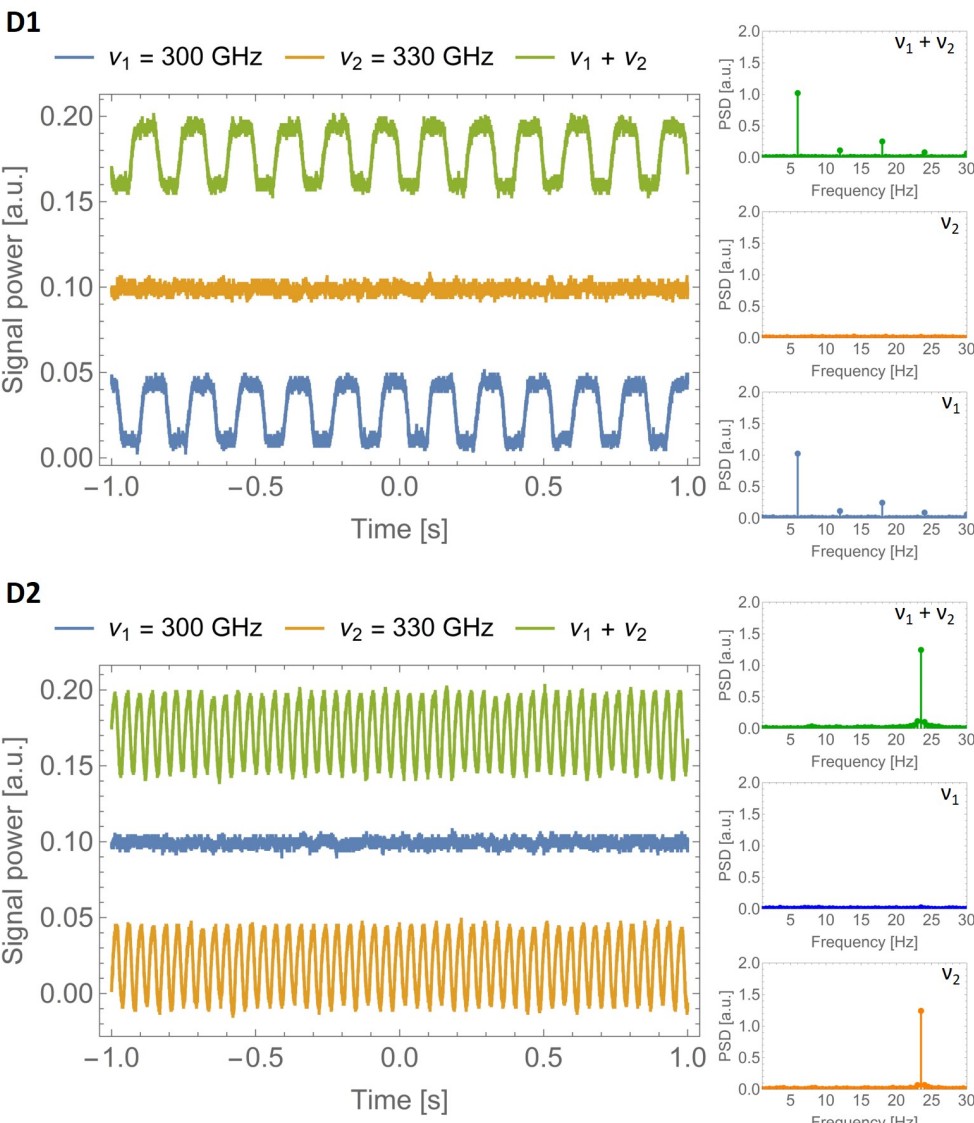

**Figure 10.** Measurements registered in two positions after the DOE—in D1 and D2. In each position, three cases have been measured: both sources are on (green), only a 300 GHz source is on (blue), and only a 330 GHz source is on (orange). The recorded intensity of signals in the time domain is plotted for both detector positions D1 (top left) and D2 (bottom left). The measurements were performed using the broadband detector. Additionally, power spectral densities of signals observed in the location D1 and D2 are illustrated on the right.

## 4. Conclusions

THz diffractive optical elements for frequency division multiplexing based on the combination of the diffraction grating and converging lens have been proposed. Four DOEs with focal lengths varying from 100 mm to 400 mm have been designed and simulated, among which two with focal lengths equal to 200 mm and 400 mm have been selected for experimental evaluation. Structures have been manufactured with an SLS-based 3D printer from PA12. The experimental data are in good agreement with the simulations and show that the designed structures allow for focus and spatially separate THz beams differing in optical frequencies. Focal spots having single millimeters in diameter have been recorded 10–25 mm apart from each other for frequencies differing by 30 GHz. It allows independent and crosstalk-free detection of these signals. An experiment showing frequency division multiplexing of two amplitude modulated signals has also been performed. It can be seen as a proof-of-concept of the frequency division multiplexing in the THz spectral range

using diffractive structures. Further work will aim at demonstrating the multichannel transmission of real digital data and measuring its parameters.

**Author Contributions:** Conceptualization, A.S. (Agnieszka Siemion); methodology, A.S. (Agnieszka Siemion) and P.K.; software, P.K.; validation, P.K., P.C. and M.S.; formal analysis, P.K., R.P. and A.S. (Agnieszka Siemion); investigation, P.C., P.Z., W.C. and M.K.; resources, P.C., W.C. and P.Z.; data curation, M.K., M.S. and A.S. (Artur Sobczyk); writing—original draft preparation, P.K., P.C., M.K. and M.S.; writing—review and editing, P.K., A.S. (Agnieszka Siemion) and R.P.; visualization, P.K., P.C., M.K., M.S. and A.S. (Artur Sobczyk); supervision, A.S. (Agnieszka Siemion); project administration, A.S. (Agnieszka Siemion). All authors have read and agreed to the published version of the manuscript.

**Funding:** Research was partially funded by the National Center for Research and Development under the Lider program (LIDER/11/0036/L-9/17/NCBR/2018).

**Data Availability Statement:** The data presented in this study are available upon request from the corresponding author. The data are not publicly available due to the qualitative nature and all transcribed data are being kept in a different form.

**Conflicts of Interest:** The authors declare no conflict of interest. The funders had no role in the design of the study; in the collection, analyses, or interpretation of data; in the writing of the manuscript, or in the decision to publish the results.

## Abbreviations

The following abbreviations are used in this manuscript:

| | |
|---|---|
| DOE | Diffractive Optical Element |
| DWL | Design Wavelength |
| MIMO | Multiple Input Multiple Output |
| SDM | Space Division Multiplexing |
| WDM | Wavelength Division Multiplexing |
| FDM | Frequency Division Multiplexing |
| SLS | Selective Laser Sintering |
| FDM* | Fused Deposition Modeling |
| DLP | Digital Light Processing |
| PP | Polypropylene |
| HIPS | High Impact Polystyrene |
| THz-TDS | THz Time Domain Spectroscopy |
| PSD | Power Spectral Density |

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
