# Peer review of "Frequency Division Multiplexing of Terahertz Waves Realized by Diffractive Optical Elements"

_applsci, doi:10.3390/app11146246_

Round 1

Reviewer 1 Report

Frequency division multiplexing of terahertz waves realized by

diffractive optical elements

PaweÅ‚ Komorowski 1* , Patrycja Czerwi ´ nska 2 , Mateusz Kaluza 2 , Mateusz Surma 2 , PrzemysÅ‚aw Zagrajek

3 , Artur Sobczyk 2 , WiesÅ‚aw Ciurapi ´ nski 3 , Ryszard Piramidowicz 1 and Agnieszka Siemion

This work by Komorowski et al describes diffractive optical elements for the frequency division multiplexing6 of terahertz waves. The structures have been designed as a combination of a binary phase grating and a converging diffractive lens. The grating allows to differentiate the frequencies, and the lens guarantees separation and focusing at the finite distance. Designed structures have been fabricated from polyamide PA12 using SLS 3D printer and experimental results are shown for different focal lengths. Focal spots having single millimeters in diameter have been recorded 10-25 mm apart from each other for frequencies differing by 30 GHz.

I can recommend the publication of this manuscript. the paper is interesting and brings really new developments.

This is some advice and comments.

  1. INTRODUCTION and abstract

  1. Material and method

What are the dielectric properties of PA12 made with a SLS 3D printer at higher frequency? They can significantly differ from raw material due to the process, and dispersion. This should be a serious limiting factor for higher frequency utilization, especially the absorption effect.

Can the authors provide a zoom picture of the realized DOE? it s  quite difficult to understand the 3D shape of the device

Can the authors give more detail how the phase distribution of the DOE is calculated and optimized?

III Experimental Results

how the authors check the beam is really collimated? What are the consequences of the multiplexing process ?

The explanation for the significant difference between theory and experiment of the f=400mm lens is not clear (line 201-204)

Please reformulate

Line 213 : why did not you perform one different XY plane like a Z scan and then quantify the different focal planes? the difference is in mm, cm  several cm range?

3.3

What are the spot sizes of the 300 & 330GHz beams ? and when they interact with the DOE ??

Please check the typos ( ex  line 294 MIMO last symbol )

Please use the same reference template  : ( ex Ref 1 ; 41 )

Reviewer 2 Report

The manuscript by Komorowski et. al. presents a very nice work about using the diffractive optical elements for the frequency division multiplexing of terahertz waves. This is a comprehensive study, including both the numerical simulations and experimental demonstrations. The manuscript is very well-written. I think it can be published in the present format.

Author Response

Dear Reviewer,

Thank you for taking time to read our manuscript.

Best regards,

Paweł Komorowski, on behalf of all co-authors